# POLICY-DRIVEN ATTACK: LEARNING TO QUERY FOR HARD-LABEL BLACK-BOX ADVERSARIAL EXAMPLES

**Ziang Yan[1,2*], Yiwen Guo[2*], Jian Liang[3], Changshui Zhang[1]**

[1] Institute for Artificial Intelligence, Tsinghua University (THUAI),
  Beijing National Research Center for Information Science and Technology (BNRist),
  Department of Automation,Tsinghua University, Beijing, P.R.China
[2] ByteDance AI Lab
[3] Alibaba Group
`yza18@mails.tsinghua.edu.cn,guoyiwen.ai@bytedance.com`
`xuelang.lj@alibaba-inc.com,zcs@mail.tsinghua.edu.cn`

## ABSTRACT

To craft black-box adversarial examples, adversaries need to query the victim model and take proper advantage of its feedback. Existing black-box attacks generally suffer from high query complexity, especially when only the top-1 decision (i.e., the hard-label prediction) of the victim model is available. In this paper, we propose a novel hard-label black-box attack named *Policy-Driven Attack*, to reduce the query complexity. Our core idea is to learn promising search directions of the adversarial examples using a well-designed policy network in a novel reinforcement learning formulation, in which the queries become more sensible. Experimental results demonstrate that our method can significantly reduce the query complexity in comparison with existing state-of-the-art hard-label black-box attacks on various image classification benchmark datasets. Code and models for reproducing our results are available at `https://github.com/ZiangYan/pda.pytorch`.

## 1 INTRODUCTION

It is widely known that deep neural networks (DNNs) are vulnerable to *adversarial examples*, which are crafted via perturbing clean examples to cause the victim model to make incorrect predictions. In a white-box setting where the adversaries have full access to the architecture and parameters of the victim model, gradients w.r.t. network inputs can be easily calculated via back-propagation, and thus first-order optimization techniques can be directly applied to craft adversarial examples in this setting (Szegedy et al., 2014; Goodfellow et al., 2015; Carlini & Wagner, 2017; Madry et al., 2018; Rony et al., 2019). However, in black-box settings, input gradients are no longer readily available since all model internals are kept secret.

Over the past few years, the community has made massive efforts in developing black-box attacks. In order to gain high attack success rates, delicate queries to the victim model are normally required. Recent methods can be roughly categorized into *score-based attacks* (Chen et al., 2017; Ilyas et al., 2018; Nitin Bhagoji et al., 2018; Ilyas et al., 2019; Yan et al., 2019; Li et al., 2020b; Tu et al., 2019; Du et al., 2019; Li et al., 2019; Bai et al., 2020) and *hard-label attacks* (a.k.a, *decision-based attacks*) (Brendel et al., 2018; Cheng et al., 2019; Dong et al., 2019; Shi et al., 2019; Brunner et al., 2019; Chen et al., 2020; Rahmati et al., 2020; Li et al., 2020a; Shi et al., 2020; Chen & Gu, 2020), based on the amount of information exposed to the adversaries from the output of victim model. When the prediction probabilities of the victim model are accessible, an intelligent adversary would generally prefer score-based attacks, while in a more practical scenario where only the top-1 class prediction is available, the adversaries will have to resort to hard-label attacks. Since less information is exposed from such feedback of the victim model, hard-label attacks often bare higher query complexity than that of score-based attacks, making their attack process costly and time intensive.

---

* The first two authors contributed equally to the work. Work was done when ZY was an intern at ByteDance AI Lab.

In this paper, we aim at reducing the query complexity of hard-label black-box attacks. We cast the problem of progressively refining the candidate adversarial example (by skillfully querying the victim model and analyzing its feedback) into a reinforcement learning formulation. At each iteration, we search along a set of chosen directions to see whether there exists any new candidate adversarial example that is perceptually more similar to its benign counterpart, i.e., in the sense of requiring less distortion. A reward is assigned to each of such search directions (treated as actions), based on the amount of distortion reduction yielded after updating the adversarial example along that direction. Such a reinforcement learning formulation enables us to learn the non-differentiable mapping from search directions to their potential of refining the current adversarial example, directly and precisely. The policy network is expected to be capable of providing the most promising search direction for updating candidate adversarial examples to reduce the required distortion of the adversarial examples from their benign counterparts. As we will show, the proposed policy network can learn from not only the queries that had been performed following the evolving policy but also peer experience from other black-box attacks. As such, it is possible to pre-train the policy network on a small number of query-reward pairs obtained from the performance log of prior attacks (with or without policy) to the same victim model. Experiments show that our policy-driven attack (PDA) can achieve significantly lower distortions than existing state-of-the-arts under the same query budgets.

## 2 RELATED WORK

In this paper, we focus on the hard-label black-box setting where only the top-1 decision of the victim model is available. Since less information (of the victim model) is exposed after each query, attacks in this category are generally required to query the victim model more times than those in the white-box or score-based settings. For example, an initial attempt named boundary attack (Brendel et al., 2018) could require ∼million queries before convergence. It proposed to start from an image that is already adversarial, and tried to reduce the distortion by walking towards the benign image along the decision boundary. Recent methods in this category focused more on gradient estimation which could provide more promising search directions, while relying only on top-1 class predictions. Ilyas et al. (2018) advocated to use NES (Wierstra et al., 2014; Salimans et al., 2017) to estimate the gradients over proxy scores, and then mounted a variant of PGD attack (Madry et al., 2018) with the estimated gradients. Towards improving the efficiency of gradient estimation, Cheng et al. (2019) and Chen et al. (2020) further introduced a continuous optimization formulation and an unbiased gradient estimation with careful error control, respectively. The gradients were estimated via issuing probe queries from a standard Gaussian distribution. To generate probes from some more powerful distributions, Dong et al. (2019) proposed to use the covariance matrix adaptation evolution strategy, while Shi et al. (2020) suggested to use customized distribution to model the sensitivity of each pixel. In contrast to these methods, our PDA proposes to use a policy network which is learned from prior intentions to advocate promising search directions to reduce the query complexity.

We note that some works also proposed to exploit DNN models to generate black-box attacks. For example, Naseer et al. (2019) used DNNs to promote the transferability of black-box attacks, while several score-based black-box attacks proposed to train DNN models for assisting the generation of queries (Li et al., 2019; Du et al., 2019; Bai et al., 2020). Our method is naturally different from them in problem settings (score-based vs hard-label) and problem formulations. In the autonomous field, Hamdi et al. (2020) proposed to formulate the generation of semantic attacks as a reinforcement learning problem to find parameters of environment (e.g., camera viewpoint) that can fool the recognition system. To the best of our knowledge, our work is the first to incorporate reinforcement learning into the black-box attacking scenario for estimating perturbation directions, and we advocate the community to consider more about this principled formulation in the future. In addition to the novel reinforcement learning formulation, we also introduce a specific architecture for the policy network which enjoys superior generalization performance, while these methods adopted off-the-shelf auto-encoding architectures.

## 3 OUR POLICY-DRIVEN ATTACK

We study the problem of attacking an image classifier in the hard-label setting. The goal of the adversaries is to perturb an benign image $x \in \mathbb{R}^n$ to fool a $k$-way victim classifier $f : \mathbb{R}^n \to \mathbb{R}^k$ into making an incorrect decision: $\arg\max_i f(x')_i \neq y$, where $x'$ is the adversarial example generated by perturbing the benign image and $y$ is the true label of $x$. The adversaries would generally prefer

adversarial examples $x'$ with smaller distortions $\|x - x'\|_2$ achieved using less queries, since these properties make the attack less suspicious and also save the cost. In this section, we first briefly review some background information that motivate our method (in Section 3.1), and then detail our reinforcement learning formulation (in Section 3.2 and Section 3.3) and the architecture of our policy network (in Section 3.4).

## 3.1 MOTIVATIONS

Most recent hard-label attacks followed a common pipeline of searching from a starting point which was already an adversarial image[1] yet not close enough to the benign one. Unlike the white-box and score-based black-box setting in which the input gradients can be calculated and used as the most effective perturbation direction, in the concerned hard-label setting, outputs of the victim model only flip on the decision boundary while keeping constant away from the boundary, making it difficult to evaluate different directions almost everywhere. In this context, the search of promising perturbation directions was restricted into the regions near the decision boundary, since these regions are arguably more informative, and binary search was used to reach the decision boundary efficiently.

Let us take a very recent attack named HopSkipJumpAttack (Chen et al., 2020) as an example. Given the current estimation $x'_s$ of the adversarial example at each iteration, HopSkipJumpAttack first performed binary search to project it onto the decision boundary of the victim model. Denote $x'$ as the updated example that was on the decision boundary already, HopSkipJumpAttack then sampled many probes around $x'$ from an isotropic Gaussian distribution, and issued these probes to the victim model as queries. The feedback of the victim model was utilized to estimate the gradient direction at $x'$, and it was updated along this direction to obtain a new estimation of the adversarial example. This process was repeated many times until the query budget was exhausted.

In comparison with boundary attack (Brendel et al., 2018), HopSkipJumpAttack was in general far more query-efficient, though a large number of queries had to be consumed for probing the local geometrics of the decision boundary of the victim model. Its superiority came from using the estimated gradient directions as the search directions, which motivated us to explore even better search directions at each iteration of the attack. As will be shown in the appendices, drawing on some geometric insights, we found that the gradient directions are in fact not the optimal search directions in the framework of HopSkipJumpAttack. We also found that the task of performing hard-label black-box attack could be naturally cast into a reinforcement learning task, thus we attempt to explore the possibility of developing a model-based method for predicting the most promising search directions for attacks. Feedbacks from the victim can provide supervision and thus the policy models in our reinforcement learning framework can be trained/fine-tuned on the fly during each attack process, such that little query is required once the model has been well-trained.

## 3.2 ATTACK AS REINFORCEMENT LEARNING PROBLEM

In this paper, we consider both targeted attacks and untargeted attacks. Given a benign example $x$, its label $y$, and the victim model $f$, an environment $\mathcal{E}(x, y, f)$ is naturally formed. The adversaries shall play the role of agent, trying to interact with the environment by issuing queries and collecting feedbacks, under a certain policy. The current example $x'_t$ on the decision boundary of the victim model (or called the candidate adversarial example) represents the state at each timestamp $t$. The agent uses a learnable policy network $g$ which will be carefully introduced in Section 3.4 to guide its actions, and the action is to update the candidate adversarial example such that less distortions are required to fool the victim model. The action here incorporates searching along a promising direction $a_t/\|a_t\|$ where $a_t \in \mathbb{R}^n$ is sampled from an isotropic Gaussian distribution whose mean vector is given by the policy network $\mu_t = g(x'_t, y, y') \in \mathbb{R}^n$ where $y'$ is the target label, and its covariance matrix is given by $\Sigma = \sigma I \in \mathbb{R}^{n \times n}$, in which the value of $\sigma \in \mathbb{R}$ is set to be gradually increased as the attack on each sample progresses, and $I \in \mathbb{R}^{n \times n}$ indicates the $n \times n$ identity matrix. With $a_t$, the agent searches along its direction $a_t/\|a_t\|$ to see whether any better candidate adversarial example can be found. For targeted attacks, the target label $y'$ is chosen by the agent from the beginning and kept unchanged during the attack process. For untargeted attacks, $x'_t$ should be on the decision boundary where one side is the ground-truth label $y$ and the other side could be regarded as the "target label" $y'$. As will be carefully introduced in Section 3.3, a reward $r_t \in \mathbb{R}$

---

[1]In practice, it is performed by randomly sampling until the adversarial constraint is satisfied, i.e., it is not classified as $y$ by the victim model, or by directly choosing a benign sample from the adversarial class.

---

**Algorithm 1** Policy-Driven Attack Algorithm

---

1: **Input:** the environment $\mathcal{E}(\boldsymbol{x}, y, f)$; the target label $y'$, initial adversarial image $\boldsymbol{x}'_1 \in \mathbb{R}^n$ which lies on the decision boundary; the policy network $g$.
2: **Output:** an adversarial example.
3: Initialize the step index $t \leftarrow 1$.
4: **while** the query count limit not reached **do**
5:      // Determine the baseline $l_t$ to evaluate the potential of different actions
6:      $\boldsymbol{\mu}_t \leftarrow g(\boldsymbol{x}'_t, y, y'), \boldsymbol{z} \leftarrow \mathrm{BS}(\boldsymbol{x}'_t + \delta \cdot \frac{\boldsymbol{\mu}_t}{\|\boldsymbol{\mu}_t\|_2}, \boldsymbol{x}, f)$, where $\mathrm{BS}(\cdot, \cdot, \cdot)$ performs binary search
7:      Set the distortion reduction of $\boldsymbol{z}$ as baseline: $l_t \leftarrow \max\{\|\boldsymbol{x}'_t - \boldsymbol{x}\|_2 - \|\boldsymbol{z} - \boldsymbol{x}\|_2, l_{\min}\}$
8:
9:      // Collect actions and rewards, and update the policy network
10:      Sample $M$ actions: $\boldsymbol{a}_{t,i} \sim \mathcal{N}(\boldsymbol{\mu}_t, \sigma_t \boldsymbol{I}), \quad i \in \{1, 2, \ldots, M\}$
11:      Assign rewards $r_{t,i}$ to each actions with our mechanism introduced in Section 3.3
12:      Update the policy network using one-step REINFORCE on $M$ pairs: $(\boldsymbol{a}_{t,i}, r_{t,i})$
13:
14:      // Update adversarial image using predicted direction
15:      $\boldsymbol{\mu}_t \leftarrow g(\boldsymbol{x}'_t, y, y')$ if $\|\boldsymbol{z} - \boldsymbol{x}\|_2 \leq \|\boldsymbol{x}'_t - \boldsymbol{x}\|_2$, otherwise $\boldsymbol{\mu}_t \leftarrow \boldsymbol{a}_{t,i^*}, i^* = \arg\max_i r_{t,i}$
16:      $\boldsymbol{x}'_{t+1} \leftarrow \mathrm{BS}(\boldsymbol{x}'_t + \epsilon \cdot \frac{\boldsymbol{\mu}_t}{\|\boldsymbol{\mu}_t\|_2}, \boldsymbol{x}, f)$
17:
18:      // Update other variables
19:      Double $\sigma_t$ if all $M$ rewards are zeros: $\sigma_{t+1} \leftarrow 2 \cdot \sigma_t$; else keep it: $\sigma_{t+1} \leftarrow \sigma_t$
20:      $t \leftarrow t + 1$
21: **end while**
22: **return** final adversarial image $\boldsymbol{x}'_t$

---

based on the performance of each action and the corresponding $\boldsymbol{a}_t$ is given to the agent for updating the parameter of the policy network. All details of our PDA are summarized in Algorithm 1.

Powered by the reinforcement learning framework, we can use policy gradient algorithms to train the policy network $g$ to generate promising search directions in a direct way. For simplicity, we use the one-step REINFORCE (Williams, 1992) in the sequel of this paper and leave the exploration of more advanced policy gradient algorithms to future work.

### 3.3 REWARD AND ACTION

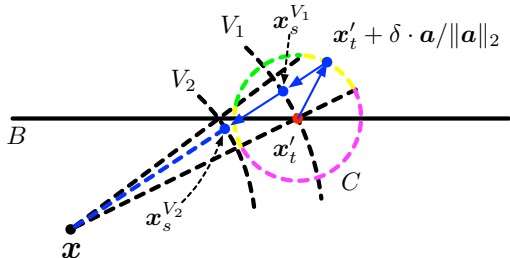

Figure 1: The reward assignment mechanism of our method. Arcs with color magenta, yellow, and green corresponded to actions with reward 0, 1, and 2, respectively.

Figure 1 illustrates how we assign the scalar reward $r_t$ given current candidate adversarial example $\boldsymbol{x}'_t$ and an action $\boldsymbol{a}_t$. The decision boundary is illustrated by a horizontal straight line (denoted by $B$) in the figure, the benign counterpart $\boldsymbol{x}$ is assumed to be below $B$, and the circle $C$ centered at $\boldsymbol{x}'_t$ with a small radius $\delta$ shows all possible locations after jumping along the directions of some actions by $\delta$ from $\boldsymbol{x}'_t$. As described earlier, the reward $r_t$ should be assigned based on the amount of potential distortion reduction brought by $\boldsymbol{a}_t$. A direct evaluation can be achieved by jumping along the direction of $\boldsymbol{a}_t$ first and then projecting the updated example back onto the decision boundary via binary search, to see how much improvement is obtained. However, since we evaluate $M$ actions $\boldsymbol{a}_{t,i}$ simultaneously at an iteration (see Algorithm 1) and binary search needs to be performed for each of them, and the overall process would be prohibitively (query-)expensive. On this point, to efficiently assess the performance of an action, we instead evaluate whether the reduction of distortion by taking

a particular action can exceed particular baselines. Concretely, we first evaluate $\boldsymbol{\mu}_t = g(\boldsymbol{x}'_t, y, y')$ as an action directly by using binary search in a way as just described. Suppose that it can reduce the required adversarial distortion[2] by $l_t$, then we setup two levels of baselines $\|\boldsymbol{x} - \boldsymbol{x}'_t\|_2 - \beta_1 \cdot l_t$ and $\|\boldsymbol{x} - \boldsymbol{x}'_t\|_2 - \beta_2 \cdot l_t$ to see whether other actions can lead to adversarial examples with closer distance (than these baselines) from the benign example $\boldsymbol{x}$, in which $\beta_1 = 0$ and $\beta_2 = 0.25$. As shown in Figure 1, for an action $\boldsymbol{a} \in \{\boldsymbol{a}_{t,i}\}$, we first obtain $\boldsymbol{x}'_t + \delta \cdot \boldsymbol{a}/\|\boldsymbol{a}\|_2$ and then move it towards $\boldsymbol{x}$ to see how much reward it can obtain. The two arcs $V_1$ and $V_2$ indicates where the same progress as the two baselines can be achieved, thus we can further project $\boldsymbol{x}'_t + \delta \cdot \boldsymbol{a}/\|\boldsymbol{a}\|_2$ onto the arcs to see if the projections (i.e. $\boldsymbol{x}_s^{V_1}$ and $\boldsymbol{x}_s^{V_2}$) are still adversarial. It can be seen that $\boldsymbol{x}_s^{V_1}$ is still adversarial yet $\boldsymbol{x}_s^{V_2}$ is not. We assign a reward 1 to such an action $\boldsymbol{a}$. If both the projections are still adversarial we shall assign a reward of 2, and if neither of them is adversarial, zero reward is assigned. Since $\boldsymbol{x}_s^{V_1}$ is not adversarial could imply that $\boldsymbol{x}_s^{V_2}$ is also not adversarial, such a way reduces the number of queries for assessing each action to at most 2 ($\boldsymbol{x}_s^{V_1}$ and $\boldsymbol{x}_s^{V_2}$) and makes our PDA more query-efficient.

### 3.4 ARCHITECTURE OF THE POLICY NETWORK

As described in Section 3.2, the goal of our policy network is to predict a direction based on which the optimal adversarial example/candidate can be easily found. Its input is the current example on the decision boundary of the victim model (together with other useful information that is available to the agent if needed) and its output is expected to be a promising search direction that shares the same dimension with its input. Naïve architecture designs of the policy network include the conventional auto-encoders and U-Net (Ronneberger et al., 2015). However, our experimental results suggest that such off-the-shelf auto-encoding architectures often offer degraded performance (see Section 4.3 for more details). We note that predicting a promising search direction is discrepant from the computer vision tasks for which these architectures are widely applied (e.g., predicting a segmentation map). Specifically, a segmentation map often aligns with the visual contents of the input image, while the promising search directions might be less correlated with the semantics of the input examples. We reckon it can be more beneficial to incorporate domain knowledge about adversarial attacks into the architecture of the policy network $g$.

On this point, we propose a new architecture for the policy network for our PDA. First, we know from HopSkipJumpAttack that the gradient direction at $\boldsymbol{x}'_t$, although not theoretically optimal, can provide strong empirical performance when serving as the search direction. Therefore, designing an architecture which could output the gradient vector at the point of its input seems to be an appropriate option for the policy network $g$. Formally, it takes the candidate adversarial example $\boldsymbol{x}'_t$, the ground-truth label $y$ of the benign example, and the target label $y'$ as input, and mapped them to a search direction in $\mathbb{R}^n$. In this spirit, the policy network is designed to own an internal classifier $h : \mathbb{R}^n \rightarrow \mathbb{R}^k$, which performs a $k$-way classification. The number $k$ can be the same as the number of prediction classes of the black-box victim model if the adversaries has such information. We hope the internal classifier $h$ can learn to distill knowledge from the victim model if possible, as such we can use the input gradient of $h$ as a descent search direction. Following the logit-diff loss developed for the white-box setting in Carlini & Wagner (2017)'s work, we propose to use:

$$g(\boldsymbol{x}'_t, y, y') = \nabla_{\boldsymbol{x}'_t} h(\boldsymbol{x}'_t)_{y'} - \nabla_{\boldsymbol{x}'_t} h(\boldsymbol{x}'_t)_y + \boldsymbol{b}, \tag{1}$$

as the output of the policy network, where a learnable bias vector $\boldsymbol{b} \in \mathbb{R}^n$ is introduced to improved the capacity and flexibility of the network. The forward process of such a policy network is basically a back-propagation process of the internal classifier $h$, and if the decision boundary of the internal classifier $h$ is aligned with that of the victim model, the output of the policy network should be the gradient direction of the victim model. Since the model is parameterized and has sufficient capacity, it can also learn to explore even better search directions other than the gradient directions.

### 3.5 (OPTIONAL) PRE-TRAINING OF THE POLICY NETWORK.

Note that the learning of the policy network can bare from large sampling complexity and may even fail to converge if its initial outputs are completely unable to cut the distortion. Just like for playing the game of go (Silver et al., 2016), we can optionally (pre-)train the policy network in a supervised manner to make the initial actions more reasonable in the reinforcement learning process, such that the samples collected in the follow-up steps are more informative and the issue can be relieved. In

---

[2]If $\boldsymbol{\mu}_t$ is unable to reduce the distortion, or $l_t < 0.05 \cdot \delta$, we clip $l_t$ to $0.05 \cdot \delta$ for numerical stability.

this section, we introduce how such pre-training can be performed for the concerned task. Recall that the goal of the policy network is to predict promising search directions for the candidate adversarial examples, thus we can construct the pre-training set $\mathbb{S}$ by collecting the intermediate results of any prior attacks to the same victim model, or we can also use a simplified policy and collect its suggested actions for constructing $\mathbb{S}$ for pre-training.

Given a small dataset $\mathbb{D} = \{(\boldsymbol{x}, y)\}$ which consists of benign examples and their ground-truth labels, it is easier to first run our PDA without learning a policy network, i.e., using an input-independent policy for it. Concretely, we can instead sample each direction $\boldsymbol{a}_t''$ from the distribution $\mathcal{N}(\boldsymbol{b}, \sigma_t \boldsymbol{I})$ at the timestamp $t$, where $\boldsymbol{b}$ here is still learnable, just like in Eq. (1). We found that such a simplified policy tends to learn a search direction that is very similar to the gradient direction. This simplified reinforcement learning problem where the policy network $g$ is now absent shows more stable training performance and each of its attack trajectories can be effectively used to pre-train the policy network, which is formulated as:

$$\mathbb{T}_{\boldsymbol{x}} \triangleq \{(\boldsymbol{x}_t'', \boldsymbol{x}, y, y_t'', \boldsymbol{a}_t'') \,|\, t \in \{1, 2, \ldots, m_{\boldsymbol{x}}\}\}, \tag{2}$$

where $m_{\boldsymbol{x}}$ is the total number of iterations, and $\boldsymbol{x}_t''$, $y_t''$, $\boldsymbol{a}_t''$ are the candidate adversarial example, the target label, and the suggested search direction by the simplified policy at each iteration, with a common timestamp $t$, respectively. To improve the compactness of the sample set for pre-training, we suggest a simple post-processing strategy to discard the tuples in $\mathbb{T}_{\boldsymbol{x}}$ with less informative candidate adversarial examples: for $i \geq 2$, we discard the $i$-th tuple in $\mathbb{T}_{\boldsymbol{x}}$ if $\|\boldsymbol{x}_i'' - \boldsymbol{x}\|_2 > 0.99\|\boldsymbol{x}_j'' - \boldsymbol{x}\|_2$, where $j$ indicates the index of any previous tuple that is decided not to be discarded. The set that contains the remaining tuples is denoted by $\mathbb{T}_{\boldsymbol{x}}^r$, and the final pre-training set $\mathbb{S}$ is constructed using this sort of set gathered from all attack trajectories using the simplified policy, i.e.,

$$\mathbb{S} = \bigcup_{(\boldsymbol{x}, y) \in \mathbb{D}} \mathbb{T}_{\boldsymbol{x}}^r. \tag{3}$$

Then it is how to pre-train the policy network given $\mathbb{S}$ for better initialization. First, according to the design of $g$ as introduced in Section 3.4, it is natural to encourage the internal classifier $h$ to perform as a classifier, and then, probably more importantly, the outputs of the policy network are encouraged to somehow align with the considered effective search directions (i.e., $\boldsymbol{a}_t''$) found by the aforementioned simplified policy. On this point, we introduce the cosine similarity $S$ together with a regularizer $\Psi$ which incorporates the classification loss to achieve these two goals, making the pre-training loss $L$ as:

$$L = \frac{1}{|\mathbb{S}|} \sum_{(\boldsymbol{x}'', \boldsymbol{x}, y, y'', \boldsymbol{a}'') \sim \mathbb{S}} -S\left(g(\boldsymbol{x}'', y, y''), \boldsymbol{a}''\right) + \lambda \cdot \Psi(h(\boldsymbol{x}), h(\boldsymbol{x}''), y, y''), \tag{4}$$

where $S(\cdot, \cdot)$ calculates the cosine similarity between its two input vectors, $\lambda$ is the coefficient for regularization, and $\Psi$ serves as a regularizer which is given by:

$$\Psi(h(\boldsymbol{x}), h(\boldsymbol{x}''), y, y'') = CE(h(\boldsymbol{x}), y) + \frac{1}{2}CE(h(\boldsymbol{x}''), y) + \frac{1}{2}CE(h(\boldsymbol{x}''), y''), \tag{5}$$

where $CE(\cdot, \cdot)$ calculates the cross-entropy loss given logits and labels. Since $\boldsymbol{x}''$ is on the decision boundary of the victim model $f$, $f$ must assign 0.5 probability to both the benign class $y$ and the adversarial class $y''$ at $\boldsymbol{x}''$, which interprets the terms $\frac{1}{2}CE(h(\boldsymbol{x}''), y)$ and $\frac{1}{2}CE(h(\boldsymbol{x}''), y'')$.

## 4 EXPERIMENTS

In this section, we evaluate the effectiveness of our method on three datasets: MNIST (LeCun et al., 2010), CIFAR-10 (Krizhevsky & Hinton, 2009), and ImageNet (Russakovsky et al., 2015). We compare our method with Boundary Attack (Brendel et al., 2018) and HopSkipJumpAttack (Chen et al., 2020) in both untargeted and targeted settings, and the required $\ell_2$ distortions given a specific query budget are evaluated, as in recent related work (Brendel et al., 2018; Chen et al., 2020; Li et al., 2020a). For a comprehensive comparison, we report the distortions at {100, 500, 1K, 5K, 10K, 25K} query budgets in all experiments in the paper. The required mean distortions for untargeted and targeted attacks are reported in Table 1 and Table 2, respectively, and the median distortions are reported in the appendices. For targeted attack, we set the target label to $y' = y + 1 \mod 10$ where $y$ is the true label for the clean image, and we only show results on MNIST and CIFAR-10 which are faster to be evaluated. All experiments are conducted on NVIDIA GTX 2080 Ti GPUs with PyTorch (Paszke et al., 2017).

Table 1: Mean $\ell_2$ distortions for performing untargeted attacks with different query budgets.

| Dataset | Victim Model | Method | @100 | @500 | @1K | @5K | 10K | @25K |
|---|---|---|---|---|---|---|---|---|
| MNIST | CNN | Boundary Attack | 10.179 | 9.970 | 9.692 | 8.960 | 3.670 | 1.850 |
| | | HopSkipJumpAttack | 7.267 | 3.576 | 2.709 | 1.843 | 1.721 | 1.641 |
| | | Ours | **2.204** | **2.053** | **1.994** | **1.674** | **1.643** | **1.639** |
| CIFAR-10 | CNN | Boundary Attack | 9.625 | 8.649 | 8.463 | 5.442 | 1.408 | 0.395 |
| | | HopSkipJumpAttack | 4.567 | 1.520 | 0.911 | 0.378 | 0.304 | **0.261** |
| | | Ours | **0.640** | **0.546** | **0.507** | **0.358** | **0.298** | 0.263 |
| | WRN | Boundary Attack | 9.268 | 8.095 | 7.972 | 3.639 | 0.664 | 0.287 |
| | | HopSkipJumpAttack | 3.041 | 1.052 | 0.649 | **0.268** | **0.213** | 0.179 |
| | | Ours | **0.700** | **0.515** | **0.453** | 0.273 | 0.217 | **0.174** |
| | ResNet50 Adv. | Boundary Attack | 10.672 | 10.273 | 10.217 | 9.064 | 4.998 | 2.378 |
| | | HopSkipJumpAttack | 7.980 | 5.453 | 4.353 | 2.521 | 2.097 | 1.793 |
| | | Ours | **2.638** | **2.454** | **2.365** | **1.926** | **1.733** | **1.570** |
| ImageNet | ResNet-18 | Boundary Attack | 65.810 | 60.861 | 60.618 | 39.359 | 13.887 | 4.877 |
| | | HopSkipJumpAttack | 36.594 | 20.881 | 14.295 | 4.846 | **2.883** | **1.479** |
| | | Ours | **11.998** | **8.200** | **6.751** | **4.093** | 2.921 | 1.481 |

Table 2: Mean $\ell_2$ distortions for performing targeted attacks with different query budgets.

| Dataset | Victim Model | Method | @100 | @500 | @1K | @5K | 10K | @25K |
|---|---|---|---|---|---|---|---|---|
| MNIST | CNN | Boundary Attack | 5.559 | 5.486 | 5.480 | 5.417 | 4.087 | 2.638 |
| | | HopSkipJumpAttack | 5.291 | 4.480 | 3.786 | 2.644 | 2.423 | **2.268** |
| | | Ours | **3.072** | **2.772** | **2.712** | **2.423** | **2.355** | 2.314 |
| CIFAR-10 | CNN | Boundary Attack | 11.194 | 10.075 | 9.381 | 7.933 | 2.778 | 0.750 |
| | | HopSkipJumpAttack | 8.864 | 3.763 | 2.118 | 0.724 | **0.549** | **0.448** |
| | | Ours | **1.426** | **1.207** | **0.930** | **0.691** | 0.589 | 0.481 |
| | WRN | Boundary Attack | 10.021 | 8.269 | 7.903 | 5.368 | 1.223 | 0.467 |
| | | HopSkipJumpAttack | 7.770 | 3.191 | 1.573 | **0.398** | **0.295** | **0.240** |
| | | Ours | **2.759** | **1.518** | **1.203** | 0.680 | 0.494 | 0.321 |
| | ResNet50 Adv. | Boundary Attack | 11.268 | 11.086 | 11.002 | 10.677 | 7.386 | 3.702 |
| | | HopSkipJumpAttack | 10.316 | 8.592 | 7.021 | 3.766 | 3.030 | 2.524 |
| | | Ours | **4.327** | **3.843** | **3.704** | **3.119** | **2.789** | **2.419** |

## 4.1 Experimental Setup

**Victim models and $h$.** For MNIST, we use the same architecture as in Carlini & Wagner (2017)'s work [3], which contains four convolutional layers and three fully connected layers and shows a test error rate of 0.59%, since it is widely adopted as a victim model in many related works. For CIFAR-10, we consider three victim models: (a) a CNN whose architecture is similar to the one considered on MNIST, which achieves a test error rate of 22.03% [4]; (b) a WRN-28-10 (Zagoruyko & Komodakis, 2016) with a test error rate of 4.03% [5]; (c) a ResNet-50 with a test error rate of 18.38% collected from the robustness package (Engstrom et al., 2019), which was adversarially trained under $\ell_2$ PGD attacks ($\epsilon = 1.0$). On ImageNet, we adopt a ResNet-18 (He et al., 2016) from the PyTorch official model zoo as the victim model, which shows a top-1 error rate of 30.24% on the ImageNet official validation set. As for the policy network $g$, we adopt a VGG-13 (Simonyan & Zisserman, 2015) architecture as its internal classifier $h$ for attacking all these victim models, i.e., the architecture of $g$ and $h$ is by no means similar to that of any of the victim models.

**Implementation details.** We mostly test our PDA with pre-training of the policy network $g$, and its performance without pre-training will be presented in the appendices. When pre-training is to be performed, we should construct a training dataset $\mathbb{S}$ for it. On MNIST and CIFAR-10, we randomly sample 5,000 images which are confirmed to be correctly classified by the victim model from their official test set to construct $\mathbb{S}$. On ImageNet, we similarly gather 50,000 images from an auxiliary dataset called ImageNetV2 and the official ImageNet validation set to construct $\mathbb{S}$. For each gathered image $\boldsymbol{x}$, we sample $m_{\boldsymbol{x}} = 500$ actions at each iteration $t$ (see Eq. (2)). When constructing $\mathbb{S}$ with the simplified policy for pre-training as described in Section 3.5, we choose the SGD optimizer without momentum, and we use a learning rate 0.003 in our PDA. Another 500 images are also collected to form the validation set for tuning all hyper-parameters for each of the three datasets (i.e., MNIST, CIFAR-10, and ImageNet).

---

[3] Pre-trained weights: `https://github.com/IBM/Autozoom-Attack`
[4] Pre-trained weights: `https://github.com/IBM/Autozoom-Attack`
[5] Pre-trained weights: `https://github.com/bearpaw/pytorch-classification`

After a pre-trained policy network is obtained, the final performance of our PDA is evaluated based on a set of 1,000 clean images disjoint from the training/validation set described in the above paragraph (i.e., these 1,000 images are not used in training the policy network and tuning hyper-parameters). At this stage, we sample only $m_{\boldsymbol{x}} = 25$ actions at each iteration for faster convergence in all experiments. We use the Adam optimizer (Kingma & Ba, 2014) with a learning rate of 0.0001 and the cross entropy regularization in Eq. 5 with a coefficient $\lambda = 0.003$ is applied. To achieve better trade-offs between exploration and exploitation, we initialize $\sigma$ in the sampling Gaussian distribution to be 0.003, and scale it at each iteration if necessary, to make sure that the ratio of the average output of the policy network and $\sigma$ lies in the range of [0.01, 0.5]. The value of $\sigma$ is doubled if all sampled actions at an iteration receive zero reward. The step size $\epsilon$ during attack is set as $0.4\|\boldsymbol{x} - \boldsymbol{x}_t'\|_2$, and the geometric regression strategy suggested by Chen et al. (2020) is also applied.

To make a fair comparison, we also sample 25 probs around each example on the decision boundary for HopSkipJumpAttack, which yields better performance than its default setting which used 100 probs. Other hyper-parameters for performing boundary attack and HopSkipJumpAttack are kept as in their original papers. The starting adversarial examples for untargeted attacks are obtained by sampling from a uniform distribution in the input space $[0, 1]^n$ until the adversarial criterion is met, and for targeted attacks we directly select a benign image from the target class as the starting point since for some victim models it is often hard to find an input from a particular class via random sampling. Once generated, the starting points are shared among all the compared attack methods.

## 4.2 COMPARISON WITH THE STATE-OF-THE-ARTS

Table 1 compares our PDA with the state-of-the-art methods for performing hard-label black-box untargeted attacks. We consider a threat model where a large number of benign examples are required to be attacked in a hard-label black-box manner, such that the queries for pre-training could be omitted. It can be easily seen that in general our method outperforms its competitors, especially in the earlier stage of attacks, which is of importance when lower query budgets are permitted. In particular, with only 100 queries, our method leads to only one-sixth to one-third distortions when compared with HopSkipJumpAttack which is the second best. With a larger query budget of 500, HopSkipJumpAttack still leads to 1.5 to 3.0 times larger distortion than our method. More interestingly, on CIFAR-10, when attacking the ResNet-50 model guarded with adversarial training, which is proved to be one of the most powerful defenses, the superiority of our PDA is in fact more significant. Such an observation is considered consistent with a phenomenon discovered in prior work (Yu et al., 2019; Zhang & Wang, 2019) which shows that adversarially trained models often have less sharp peaks and cliffs on the decision boundary, making it easier for our policy network to capture. Table 2 shows the results for targeted attacks and our PDA again outperforms others in general. Performance of our PDA under different pre-training configurations is given in the appendices. In practice, pre-training is recommended since it is crucial for the superior performance of our method.

## 4.3 ABLATION STUDY

Table 3: Comparison of choosing different architectures for the policy network. Mean $\ell_2$ distortions for performing untargeted attacks are evaluated.

| Architecture | @100 | @500 | @1K | @5K | @10K | @25K |
|---|---|---|---|---|---|---|
| U-Net (small) | 1.023 | 0.791 | 0.613 | 0.374 | 0.300 | 0.276 |
| U-Net (medium) | 0.871 | 0.772 | 0.587 | 0.362 | **0.288** | 0.273 |
| U-Net (large) | 0.882 | 0.741 | 0.589 | 0.365 | 0.308 | **0.262** |
| Ours | **0.640** | **0.546** | **0.507** | **0.358** | 0.298 | 0.263 |

We perform an ablation study on how the architecture design of the policy network would affect the performance of our PDA. More specifically, we train policy networks with several different architectures on the same set $\mathbb{S}$ and then attempt to attack the CNN victim model on CIFAR-10 using these policy networks. In addition to our design (as introduced in Section 3.4) which leverages the gradient of an internal classifier $h$, we mostly consider the U-Net (Ronneberger et al., 2015) which is a popular option for the learning models for assisting adversarial attacks. We compare our proposed architecture with several U-Nets with different configurations. Details of their architectures can be found at `https://github.com/milesial/Pytorch-UNet`. Note that, for U-Net models, the cross-entropy regularization is not applied, since they do not perform classification and there

is no logit for computing such a loss. The performance of different policy networks in terms of the mean $l_2$ distortion is reported in Table 3. Obviously, we can see that the proposed architecture outperforms U-Net significantly in the framework of our PDA.

## 5 CONCLUSION

Existing hard-label black-box attacks often suffer from very high query complexity. In this paper, we have introduced a model-based method (i.e., PDA) for learning from past queries and model feedbacks, based on a reinforcement learning formulation of the attack. We have developed a novel architecture for the policy network that is designed to suggest promising search directions for the adversarial examples. Moreover, it has been demonstrated that pre-training of such a policy network, which is crucial for the attack performance, can be effectively performed using prior attacking logs on the same victim model. Experimental results on various victim models (including both naturally and adversarially trained ones) trained on different datasets (including MNIST, CIFAR-10, and ImageNet) suggest that the proposed PDA significantly outperforms existing state-of-the-arts in terms of query efficiency.

ACKNOWLEDGMENTS

This work is funded by the National Key Research and Development Program of China (No. 2018AAA0100701) and the NSFC 61876095.

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

## A  OPTIMAL SEARCH DIRECTION FOR HOPSKIPJUMPATTACK

We illustrate the optimal search direction for HopSkipJumpAttack in a two-dimensional input space in Figure 2. The decision boundary is illustrated by a horizontal straight line (denoted by $B$), the benign counterpart $\boldsymbol{x}$ of the adversarial example is assumed to be below $B$, and the circle $C$ centered at the candidate adversarial example $\boldsymbol{x}'_t$ with a small radius $\delta$ shows all possible locations after jumping along some directions by distance $\delta$ from $\boldsymbol{x}'_t$. The gradient direction $\boldsymbol{u}_g$ shall be vertical under the locally linear assumption, and after updating $\boldsymbol{x}'_t$ along that direction (by $\delta$) and projecting the updated image back onto the decision boundary $B$ (path marked in blue), we obtain $\boldsymbol{x}^g_t$. Let the straight line $T$ be the tangent line of $C$ which goes through $\boldsymbol{x}$, and $\boldsymbol{u}_o$ is the direction which is perpendicular to $T$. If we update $\boldsymbol{x}'_t$ along $\boldsymbol{u}_o$ and then project the result back to $B$ (path marked in green), $\boldsymbol{x}^o_t$ is obtained. Clearly, $\boldsymbol{x}^o_t$ would have a smaller distortion than $\boldsymbol{x}^g_t$: $\|\boldsymbol{x}^o_t - \boldsymbol{x}\|_2 < \|\boldsymbol{x}^g_t - \boldsymbol{x}\|_2$, indicating $\boldsymbol{u}_o$ is a better direction than the gradient $\boldsymbol{u}_g$ in the sense of cutting distortion. It is also easy to verify $\boldsymbol{u}_o$ is the optimal updating direction in this two-dimensional case.

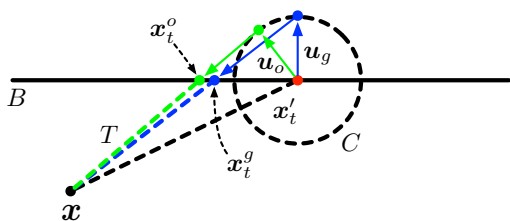

Figure 2: The optimal search direction for HopSkipJumpAttack.

# B MEDIAN DISTORTIONS

We report the median $l_2$ distortions over perturbing test images at different query count budgets for performing untargeted attacks in Table 4. It can be easily seen that our method outperforms existing state-of-the-arts on all test cases, especially in the earlier stages of the attacks.

Table 4: Median $\ell_2$ distortions for performing targeted attacks with different query budgets.

| Dataset | Victim Model | Method | @100 | @500 | @1K | @5K | 10K | @25K |
|---|---|---|---|---|---|---|---|---|
| MNIST | CNN | Boundary Attack | 10.108 | 9.946 | 9.662 | 9.098 | 3.604 | 1.825 |
| | | HopSkipJumpAttack | 7.114 | 3.505 | 2.664 | 1.813 | 1.693 | 1.616 |
| | | Ours | **2.165** | **2.024** | **1.955** | **1.631** | **1.602** | **1.595** |
| CIFAR-10 | CNN | Boundary Attack | 8.825 | 7.985 | 7.970 | 5.000 | 1.189 | 0.332 |
| | | HopSkipJumpAttack | 3.629 | 1.188 | 0.739 | 0.321 | 0.262 | 0.227 |
| | | Ours | **0.543** | **0.463** | **0.429** | **0.298** | **0.249** | **0.218** |
| | WRN | Boundary Attack | 8.740 | 7.778 | 7.765 | 3.456 | 0.607 | 0.263 |
| | | HopSkipJumpAttack | 2.638 | 0.927 | 0.589 | 0.248 | 0.199 | 0.168 |
| | | Ours | **0.578** | **0.448** | **0.400** | **0.244** | **0.195** | **0.158** |
| | ResNet50 Adv. | Boundary Attack | 9.303 | 8.881 | 8.869 | 8.647 | 5.071 | 2.296 |
| | | HopSkipJumpAttack | 7.569 | 5.439 | 4.362 | 2.424 | 1.972 | 1.672 |
| | | Ours | **2.545** | **2.396** | **2.301** | **1.806** | **1.624** | **1.464** |
| ImageNet | ResNet-18 | Boundary Attack | 64.303 | 58.947 | 58.888 | 37.352 | 11.123 | 3.299 |
| | | HopSkipJumpAttack | 34.955 | 17.309 | 11.057 | 3.309 | 1.993 | 1.046 |
| | | Ours | **10.751** | **7.522** | **5.554** | **2.997** | **1.877** | **0.998** |

# C EFFECTS OF PRE-TRAINING

In this section, we study the effects of pre-training in our PDA. We only show results on MNIST with CNN as the victim model which is faster to be evaluated. We test our PDA under several different pre-training configurations: (a): the policy network does not own the internal classifier $h$, i.e., its output is input-agnostic and thus pre-training is meaningless in this case; (b): the policy network has a VGG-13 as its internal classifier as in the main paper, and it is pre-trained on data sets with smaller sizes. To achieve the goal, we first sample a subset $\mathbb{D}'$ with size $|\mathbb{D}'| \in \{0, 50, 500, 5000\}$ from $\mathbb{D}$ which has 5,000 benign images in total and is used to create $\mathbb{S}$ in the main paper, and then we collect tuples from $\mathbb{D}'$ to form a possibly smaller pre-training set.

Table 5: Comparison of different pre-training configurations for the policy network. Mean $\ell_2$ distortions for performing untargeted attacks are evaluated.

| Method | $\|\mathbb{D}'\|$ | Has $h$? | @100 | @500 | @1K | @5K | 10K | @25K |
|---|---|---|---|---|---|---|---|---|
| Boundary Attack | - | - | 10.179 | 9.970 | 9.692 | 8.960 | 3.670 | 1.850 |
| HopSkipJumpAttack | - | - | 7.267 | 3.576 | 2.709 | 1.843 | 1.721 | 1.641 |
| Ours | - | No | 7.314 | 3.489 | 2.981 | 1.809 | 1.772 | 1.648 |
| Ours | 0 | Yes | 9.052 | 6.438 | 4.994 | 2.130 | 1.720 | 1.665 |
| Ours | 50 | Yes | 3.028 | 2.571 | 2.320 | 1.696 | 1.636 | 1.632 |
| Ours | 500 | Yes | 2.287 | 2.104 | 2.024 | **1.668** | **1.629** | **1.625** |
| Ours | 5,000 | Yes | **2.204** | **2.053** | **1.994** | 1.674 | 1.643 | 1.639 |

Table 5 summarizes our results, in which the second column indicates the size of $\mathbb{D}'$ which reflects the cost for pre-training, in particular, $|\mathbb{D}'| = 0$ means no pre-training is applied, and the third column indicates whether or not the policy network owns an internal classifier. We see when pre-training is not applied, with a simplified policy (i.e., without $h$) our PDA can achieve comparable performance to HopSkipJumpAttack, yet directly incorporating a randomly initialized internal classifier into the policy network would lead to much worse result since, in this case, the initial search directions for adversarial example suggested by the policy network are nearly random and often completely failed to reduce the required distortions. Moreover, it can also be seen that, with the size of the pre-training set increased, the performance of our PDA is also gradually improved. Figure 3 provides visualizations of generated adversarial examples on a randomly selected benign image. Each row in the figure represents a pre-training configuration, and images in each column are candidate adversarial examples under a certain query count budget. The benign example is classified as "7" by the victim model, and all images in the figure are classified as "3". The first column of the

figure shows the common initial adversarial example for all configurations, which is generated via sampling from the uniform distribution in $[0, 1]^n$ as described in the main paper. It can be seen that our method could provide adversarial images with higher qualities, especially in the early stage.

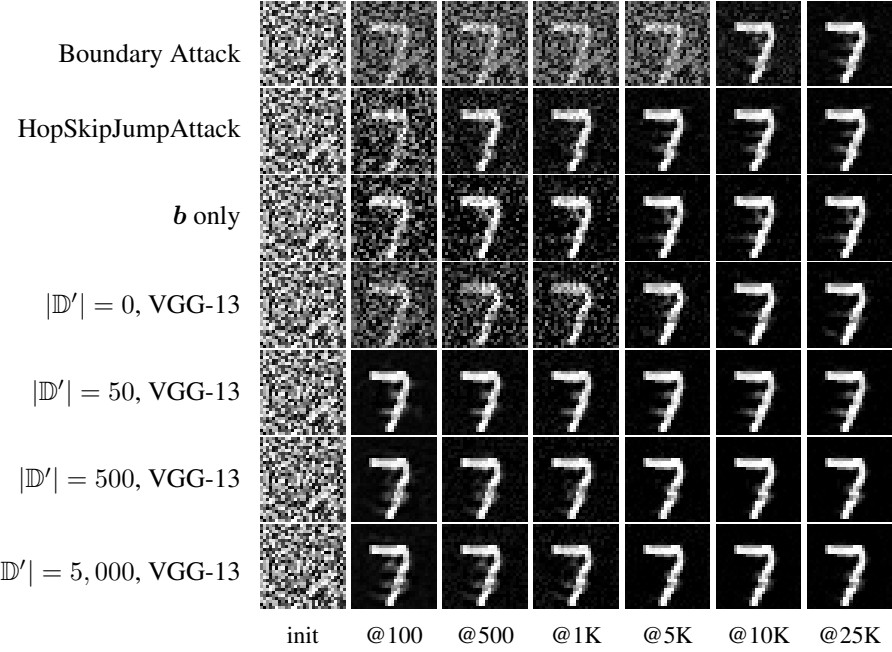

Figure 3: Visualization of generated adversarial images using different pre-training configurations. All shown images are classified as "3" by the victim model.

Since favorable results can be obtained with light or even no pre-training of the policy network, our PDA is also applicable to scenarios where only a few adversarial examples are to be generated. This paper introduces a novel perspective of viewing hard-label black-box attacks (as a reinforcement learning problem), thus more advanced policy gradient methods an also be tested for improving the performance of our PDA, with or without pre-training.

## D  VALUES OF BASELINE LEVELS

In this section, we explain how the values of $\beta_1 = 0$ and $\beta_2 = 0.25$ are selected. These values are tuned under a constraint of $\beta_1 < \beta_2$. We try different combinations of values in $\{0, 0.25, 0.5, 1.0\}$ for $\beta_1$ and $\beta_2$ on MNIST and found that $\beta_1 = 0$, $\beta_2 = 0.25$ provide the best result on the validation set (containing 500 images as described in Section 4.1), and these values are then applied to all other experiments. In Table 6 we report the mean $l_2$ distortions when attacking a CNN victim on MNIST test images (1,000 images as described in Section 4.1) at different query count budgets. We see that the performance of our method is fairly robust to tested values of $\beta_1$ and $\beta_2$, and smaller values usually have better performance on the late stage of attacks.

Table 6: Comparison of choosing different $\beta_1$ and $\beta_2$. Mean $\ell_2$ distortions for performing untargeted attacks on MNIST are evaluated, and the victim model is CNN.

| $(\beta_1, \beta_2)$ | @100 | @500 | @1K | @5K | @10K | @25K |
|---|---|---|---|---|---|---|
| (0, 0.25) | 2.204 | 2.053 | 1.994 | 1.674 | **1.643** | **1.639** |
| (0, 0.5) | 2.202 | 2.035 | **1.973** | **1.673** | 1.648 | 1.641 |
| (0, 1.0) | 2.204 | 2.049 | 1.993 | 1.688 | 1.650 | 1.642 |
| (0.25, 0.5) | 2.202 | 2.032 | 1.977 | 1.782 | 1.767 | 1.748 |
| (0.25, 1.0) | 2.203 | 2.038 | 1.982 | 1.781 | 1.766 | 1.748 |
| (0.5, 1.0) | **2.199** | **2.028** | 1.986 | 1.902 | 1.885 | 1.859 |

# E  TRANSFERABILITY OF THE PRE-TRAINED POLICY NETWORK

In this section, we study the transferability of the pre-trained policy network across different victim models on the same dataset. To do so, on CIFAR-10, we use policy networks pre-trained on {CNN, WRN, ResNet-50 Adv.} to attack all these three victim models. Table 7 summarizes our results, in which the first column is the victim model used to evaluate the attack performance, and the second column is the victim model used to collect dataset $\mathbb{S}$ for pre-training. We see from Table 7 that when using a policy network pre-trained on a different victim model to attack, the attacking performance is degraded on all test cases. However, in the early stage of attacking process, our PDA can still consistently provide smaller distortions even when the policy network is pre-trained on a different victim model, enabling attackers to benefit from our PDA by using it to provide high quality starting points for other attacks. More importantly, the transferability allows one to pre-train the policy network on some local models, thus the queries consumed in collecting the pre-training dataset can be practically saved.

Table 7: Transferability of pre-trained policy networks on CIFAR-10. Last six columns are median $\ell_2$ distortions for performing untargeted attacks with different query budgets.

| Victim Model for Evaluation | Victim Model for Pre-Training | @100 | @500 | @1K | @5K | 10K | @25K |
|---|---|---|---|---|---|---|---|
| | CNN | **0.640** | **0.546** | **0.507** | **0.358** | **0.298** | 0.263 |
| CNN | WRN | 2.719 | 1.683 | 1.322 | 0.565 | 0.399 | 0.278 |
| | ResNet50 Adv. | 1.405 | 1.087 | 0.895 | 0.422 | 0.315 | **0.249** |
| | CNN | 2.060 | 1.299 | 1.035 | 0.471 | 0.325 | 0.216 |
| WRN | WRN | **0.700** | **0.515** | **0.453** | **0.273** | **0.217** | **0.174** |
| | ResNet50 Adv. | 2.028 | 1.474 | 1.173 | 0.482 | 0.309 | 0.197 |
| | CNN | 6.324 | 5.257 | 4.790 | 3.146 | 2.449 | 1.827 |
| ResNet50 Adv. | WRN | 6.737 | 5.697 | 5.203 | 3.524 | 2.807 | 2.020 |
| | ResNet50 Adv. | **2.638** | **2.454** | **2.365** | **1.926** | **1.733** | **1.570** |

# F  ATTACK PERFORMANCE ON THE PRE-TRAINING DATASET

The untargeted attack performance on the pre-training dataset $\mathbb{S}$ is reported in Table 8. We see our method has the best performance on the pre-training images, just like in the test images. By comparing the performance of our method on the pre-training images and the test images, we see the "overfitting" of our method is moderate and acceptable in practice.

Table 8: Mean $\ell_2$ distortions for performing untargeted attacks with different query budgets on the pre-training dataset $\mathbb{S}$.

| Dataset | Victim Model | Method | @100 | @500 | @1K | @5K | 10K | @25K |
|---|---|---|---|---|---|---|---|---|
| MNIST | CNN | Boundary Attack | 10.252 | 10.067 | 9.774 | 9.059 | 3.700 | 1.871 |
| | | HopSkipJumpAttack | 7.373 | 3.607 | 2.728 | 1.855 | 1.732 | **1.651** |
| | | Ours | **2.229** | **2.075** | **2.015** | **1.692** | **1.660** | 1.654 |
| CIFAR-10 | CNN | Boundary Attack | 9.558 | 8.578 | 8.385 | 5.462 | 1.403 | 0.384 |
| | | HopSkipJumpAttack | 4.396 | 1.448 | 0.874 | 0.371 | 0.300 | 0.258 |
| | | Ours | **0.581** | **0.494** | **0.464** | **0.343** | **0.290** | **0.257** |
| | WRN | Boundary Attack | 9.318 | 8.047 | 7.928 | 3.567 | 0.663 | 0.293 |
| | | HopSkipJumpAttack | 3.020 | 1.050 | 0.655 | **0.273** | **0.218** | 0.184 |
| | | Ours | **0.660** | **0.506** | **0.448** | 0.274 | 0.221 | **0.177** |
| | ResNet50 Adv. | Boundary Attack | 10.481 | 10.091 | 10.039 | 8.990 | 5.020 | 2.422 |
| | | HopSkipJumpAttack | 7.938 | 5.473 | 4.390 | 2.575 | 2.137 | 1.817 |
| | | Ours | **2.531** | **2.377** | **2.327** | **1.981** | **1.791** | **1.616** |
| ImageNet | ResNet-18 | Boundary Attack | 66.007 | 61.033 | 60.767 | 39.710 | 13.548 | 4.566 |
| | | HopSkipJumpAttack | 36.812 | 20.701 | 14.146 | 4.779 | 2.815 | 1.440 |
| | | Ours | **8.132** | **6.900** | **5.027** | **3.761** | **2.148** | **1.361** |

# G  COMPUTATION AND MEMORY COMPLEXITY

As a policy network is involved in our proposed attack process, our PDA method naturally has higher computation and memory complexity than baseline methods. In our experiment for attacking a ResNet-18 model on ImageNet, for each 100 queries to the victim model, it costs our method extra 947ms on a single GPU (excluding the inference time of the victim model) to fine-tune the policy

network, while HopSkipJumpAttack requires 354ms. Although it seems that our method is computational more intensive, the extra overhead is acceptable considering that the inference of the victim model is also costly (672ms), also the run-time of our method can be reduced by using multiple GPUs. As for GPU memory consumption, when attacking the ResNet-18 victim using batch size 25, our method needs additional ∼5GB GPU memory compared with HopSkipJumpAttack which needs ∼2GB GPU memory. The computational and memory cost of our method can be reduced by compressing the policy network, which is to be explored in future work.

## H   COMPARISON TO SIGN-OPT

In this section, we compare our method to a recent hard-label black-box attack named Sign-OPT (Cheng et al., 2020). We directly use their official implementation and hyper-parameters [6]. When evaluating Sign-OPT, we use the same victim networks and initial adversarial images as in our method to make a fair comparison. The untargeted attack performance of Sign-OPT on MNIST and CIFAR-10 is reported in Table 9. We see, in general, our method outperforms Sign-OPT, especially in the earlier stage of attack.

Table 9: Mean $\ell_2$ distortions for performing untargeted attacks with different query budgets.

| Dataset | Victim Model | Method | @100 | @500 | @1K | @5K | 10K | @25K |
|---|---|---|---|---|---|---|---|---|
| MNIST | CNN | Boundary Attack | 10.179 | 9.970 | 9.692 | 8.960 | 3.670 | 1.850 |
| | | HopSkipJumpAttack | 7.267 | 3.576 | 2.709 | 1.843 | 1.721 | 1.641 |
| | | Sign-OPT | 10.009 | 5.398 | 3.381 | 2.019 | 1.698 | 1.669 |
| | | Ours | **2.204** | **2.053** | **1.994** | **1.674** | **1.643** | **1.639** |
| CIFAR-10 | CNN | Boundary Attack | 9.625 | 8.649 | 8.463 | 5.442 | 1.408 | 0.395 |
| | | HopSkipJumpAttack | 4.567 | 1.520 | 0.911 | 0.378 | 0.304 | 0.261 |
| | | Sign-OPT | 6.018 | 3.221 | 1.498 | 0.404 | **0.287** | **0.250** |
| | | Ours | **0.640** | **0.546** | **0.507** | **0.358** | 0.298 | 0.263 |
| | WRN | Boundary Attack | 9.268 | 8.095 | 7.972 | 3.639 | 0.664 | 0.287 |
| | | HopSkipJumpAttack | 3.041 | 1.052 | 0.649 | **0.268** | **0.213** | 0.179 |
| | | Sign-OPT | 5.113 | 2.099 | 0.757 | 0.341 | 0.226 | 0.193 |
| | | Ours | **0.700** | **0.515** | **0.453** | 0.273 | 0.217 | **0.174** |
| | ResNet50 Adv. | Boundary Attack | 10.672 | 10.273 | 10.217 | 9.064 | 4.998 | 2.378 |
| | | HopSkipJumpAttack | 7.980 | 5.453 | 4.353 | 2.521 | 2.097 | 1.793 |
| | | Sign-OPT | 9.211 | 7.840 | 4.101 | 2.059 | 1.784 | 1.649 |
| | | Ours | **2.638** | **2.454** | **2.365** | **1.926** | **1.733** | **1.570** |

---

[6]https://github.com/cmhcbb/attackbox

