# OpenReview forum: "Policy-Driven Attack: Learning to Query for Hard-label Black-box Adversarial Examples"
_ICLR.cc/2021/Conference — ICLR 2021 Poster_

### Official Review · AnonReviewer2 · 2020-10-27
**Official Blind Review #2**

**Rating:** 7
**Confidence:** 5

**Review:**

This paper proposes a new hard-label black-box adversarial attack method based on reinforcement learning. The general idea is to improve the adversarial noise compressing efficiency taking advantage of past queries and the policy network. Experiments are conducted on MNIST, CIFAR-10 and ImageNet and achieved superior performance compared to other hard-label black-box attacks. In addition, ablation studies are conducted to verify the effectiveness of the proposed attack.

There is almost no intersection between reinforcement learning and adversarial attacks for three main reasons. First of all, the action space of optimizing adversarial noises is too large for the reinforcement learning algorithm to converge. Secondly, the optimal attack strategies of different images are quite different, so it is difficult to learning a unified policy network to attack the whole image classification dataset. In addition, it is difficult to design the value function to guide the attack process. This paper innovatively applies reinforcement learning in the neighborhood of the decision boundary and solves the above problem, realizing the adaptive adjustment of the noise-searching direction.

I have three questions about the practicality of the proposed attack. First, according to algorithm 1, the single action in the policy-driven attack is a direction sampled from Gaussian distribution, so the dimensionality of the action space is still high. Will this affect the convergence of the policy network? Secondly, if the policy network is optimized after each query, will the attack speed be severely lagged? In addition, section 3.5 of this paper discussed the pre-training of the policy network. I wonder whether the policy and the target model are coupled. If the target model changes, is it necessary to retrain the policy network? This may affect the practicability of the attack.

---

> ### Author Response · Authors · 2020-11-20
> **Response to AnonReviewer2**
>
> We would like to thank the reviewer for the positive feedback on the contribution of our work. Below are our response to the comments:
> * To the question about the dimensionality of the action space and convergence of the policy network, we would like to explain in two aspects:
>     * Although the dimensionality of the action space is high, in our implementation, the sampled actions are practically not too far away (in the sense of angular distance) from the output direction of the policy network. For example, on CIFAR-10 where the dimensionality of the action space is 3x32x32=3072, after adjusting the relative scale of the mean and std of the normal distribution as described in Section 4.1, most sampled actions (>99.9%) would have cosine similarities between 0.6 and 0.7 with the output of the policy network. Thus if the policy network could be reasonably initialized (for example via pre-training), in practice one can expect the agent to behave not too poorly in the environment.
>     * We have leveraged plenty of domain knowledge to reduce the sampling space significantly, by using, for instance, binary search to facilitate the attack.
> * To the question about attack speed, we explain that, for attacking a ResNet-18 model on ImageNet, although it costs our method extra 947ms to obtain every 100 queries and fine-tune the policy network, the inference of the victim model is also costly (672ms). The run-time of our method can be largely reduced by compressing the policy network, which is to be explored in future work, or by training/fine-tuning on multiple GPUs. We’ve added a section (Appendix F) in the revised paper to discuss the computation overhead of our method.
> * We appreciate the insightful question about the necessity of policy network retraining. In our method, the target model is the core part of the environment in which the agent plays. However, even if the target model changes, due to the transferability we could still benefit from the old policy network which is pre-trained on the old target model, although the performance is inferior to the case of retraining the policy network. We’ve added a section (Appendix E) in the revised paper to discuss the transferability.

---

### Official Review · AnonReviewer4 · 2020-10-28
**The idea is good, but evaluation is biased**

**Rating:** 6
**Confidence:** 4

**Review:**

Summary:
This paper proposes a new hard-label black-box adversarial attack method based on reinforcement learning. The authors formulate the black-box attacking problem as a reinforcement learning problem, and design a policy network to learn the appropriate attack directions, in order to achieve more efficient attacks. The proposed policy-driven attack (PDA) algorithm is able to craft adversarial examples with lower distortions under the same query budgets. Experiment results show that with a pre-trained policy model, PDA outperforms two baseline methods, Boundary Attack and HopSkipJumpAttack.

Strengths:
1. This paper is the first to incorporate reinforcement learning into black-box attacking. The proposed reinforcement learning formulation and the architecture of policy network are interesting and novel.
2. A significant amount of experiments are done to show the effectiveness and efficiency of the proposed policy-driven attack compared with baselines. Moreover, the ablation experiment in the main paper and appendix provide useful insights of how every component of PDA works.
3. The paper is in general well-written and easy to follow. Implementation details are clearly described.


Weakness:
1. My main concern is that there is a mismatch between the claim and the experiment. In the experiments, the paper mainly shows the results with pre-trained policy networks. However, the pertaining dataset S requires a lot of queries to the victim, which is conflictive with the purpose to reduce the number of queries. Although the authors mention that they consider the case where a large number of adversarial examples are required so that the pre-training samples can be omitted, the experiment setting does not match this assumption (500 queries * 5000 images in the pretraining dataset, versus 25 queries * 1000 images in the testing dataset which is used to evaluate the performance). It is not clear whether the performance of PDA is still the best when the number of adversarial examples keeps increasing, as it is a deep RL network, not guaranteed to be stable. ———— The authors may consider the following ways to make the experiment more convincing: (1) using much more images in the testing phase than in the pertaining phase; (2) sampling the pre-training dataset from another classifier on the same dataset, rather than from the real victim (in this way the queries to the victim can be really saved); (3) incorporating both the pre-training dataset and the testing dataset in the final performance evaluation.
2. As also pointed out by the authors, reinforcement learning is itself sample-consuming. Applying RL to a complicated task could be difficult and non-stable. This paper proposes a few methods to facilitate learning, e.g.,  domain knowledge, pre-training, which are nice. But according to table 5, without pre-training, PDA performs similarly to HopSkipJumpAttack (even worse when internal classifier h is used), while the latter is easier to implement. This is a little concerning, suggesting the necessarily of pre-training for PDA if one wants to benefit from using PDA. I might be wrong, but if this is the case, it will be better for the authors to emphasize the importance of pre-training, and show the reasonability of using pre-training samples as stated in point 1 above.

Minor comments:
- Although the authors formulate the attacking process as a decision making problem and use a policy network to generate actions / search directions, I do not see the necessity of using RL techniques. The main reason is, there is no “horizon”, as the agent does not consider future rewards. (I am not sure whether the authors consider the future rewards in the implementation, but the description seems to only consider the current state.) Thus it might be enough to formalize this problem as a contextual bandit. This point does not influence my rating though.

---

> ### Author Response · Authors · 2020-11-20
> **Response to AnonReviewer4**
>
> We would like to thank the reviewer for recognizing the contributions of our work and providing valuable feedback, and we have carefully revised the paper following the suggestions.
> * We appreciate the constructive suggestions for experiments.
>     * We have conducted an experiment on MNIST to further verify the stability of our method where there are more images in the testing phase. We used 500 images to pre-train the policy network and varied the number of test images from 1000 to 8000. The below table shows the mean l2 distortion on test images for untargeted attacks with different query budgets, and we can see from the table that our method is statistically stable as the scale of the test set grows:
> | #Test images | Method | @100 | @500 | @1K | @5K | @10K | @25K |
> |--|--|--|--|--|--|--|--|
> | 1000 | Boundary Attack | 10.179 | 9.970 | 9.692 | 8.960 | 3.670 | 1.850 |
> | 1000 | HopSkipJumpAttack | 7.267 | 3.576 | 2.709 | 1.843 | 1.721 | 1.641 |
> | 1000 | Ours | 2.287 | 2.104 | 2.024 | 1.668 | 1.629 | 1.625 |
> | 2000 | Boundary Attack | 10.272 | 10.085 | 9.801 | 9.081 | 3.792 | 1.870 |
> | 2000 | HopSkipJumpAttack | 7.366 | 3.593 | 2.713 | 1.847 | 1.726 | 1.645 |
> | 2000 | Ours | 2.305 | 2.117 | 2.037 | 1.683 | 1.648 | 1.643 |
> | 4000 | Boundary Attack | 10.238 | 10.060 | 9.782 | 9.065 | 3.792 | 1.877 |
> | 4000 | HopSkipJumpAttack | 7.343 | 3.595 | 2.719 | 1.849 | 1.728 | 1.647 |
> | 4000 | Ours | 2.308 | 2.117 | 2.038 | 1.682 | 1.647 | 1.642 |
> | 8000 | Boundary Attack | 10.262 | 10.080 | 9.799 | 9.080 | 3.777 | 1.877 |
> | 8000 | HopSkipJumpAttack | 7.361 | 3.617 | 2.737 | 1.862 | 1.741 | 1.659 |
> | 8000 | Ours | 2.317 | 2.128 | 2.049 | 1.691 | 1.654 | 1.651 |
>
>    * We have conducted an experiment to study the performance of the policy network pre-trained on one victim model in attacking other victim models. See Appendix E for details. Under such circumstances, the attacker can still benefit from our method.
>     * We have also followed the suggestion and tested the attack performance on the pre-training images in Appendix G. Similar observations can be made that our method led to much lower distortions in comparison with other methods in the early stage of the attack.
> * To the comment about importance of pre-training, we would like to explain that indeed pre-training is crucial for achieving the optimal experimental performance of our method, which could possibly be ascribed to the fact that the sampling complexity is still high in the dimensional sampling space of the hard-label black-box attack. Towards accelerating the sampling process, we have made several efforts including introducing more domain knowledge and a more suitable policy network, as have been recognized by the reviewer. We have revised Section 4.2 and Section 5 to emphasize the importance of pre-training in the paper.
> * We explain that, in our current implementation, future rewards are not incorporated. However, in principle, we believe that future rewards should be helpful to the problem, and we will study it more in our future work.

---

> ### Comment · AnonReviewer4 · 2020-11-23
> **The additional experiment addressed my main concern**
>
> The additional experimental result looks good and supports the claim of the paper. Considering other strengths of the paper, I would like to increase my score to 6.

---

### Official Review · AnonReviewer1 · 2020-10-29
**promising new way of blackbox attacks**

**Rating:** 7
**Confidence:** 3

**Review:**

**Summary** :
This work proposes to formulate the problem of black-box adversarial attacks as learning a policy network that predicts offsets to some initial guesses of adversarial examples. The proposed PDA method tackles the problematic situation of queries with only hard labels. A specific reward, architecture, and pretraining setups are designed for the policy network to learn useful directions in updating the adversarial examples. Experiments on MNIST, CIFAR10, and ImageNet show the proposed PDA produced less distorted adversarial examples with less sampling budget for the black box attacks than two recent baseline attacks.

**Strengths** :
- The quarry efficiency is demonstrated compared to the two baselines (Boundary Attack, HopSkipJumpAttack) with less distortion on standard datasets and models.
- The paper is well written and well presented. The mathematical notations are clear and informative.
- The use of policy networks to produce black-box adversarial attacks is a novel idea. It might open a direction for other fields interested in security besides the usefulness in the adversarial ML community.

**Weaknesses**:
- The work has a limited ablation study. While the authors did provide an ablation on the policy network architecture and pretraining, some essential components of the reward setup and the algorithm are given without proper justification nor ablation study, e.g., the 0.5 and 0 of $\beta$ values in the reward limits in Algorithm 1. It would also be appreciated to show the attack's transferability between different models on the same datasets.
- The proposed pipeline is complicated compared to the baselines. Pre-Training a policy network, then training the network adds to the computing complexity of the problem. While the sampling efficiency has increased, the compute and memory efficiency of the attacks have decreased. The baselines NATTACK: ( Li et. al , ICML 2019) and AutoZoom (Tu et. al, AAAI 2019 ) could have been adapted and compared against in the paper to see how useful are the policy networks.
- Missing some references[a,b]. [a] uses a deep U-net network to produce adversarial attacks that are black box-transferable to unseen models. It bears similarity to the proposed network architecture in which the goal is to produce differences rather than predict the perturbation directly. The work in [b] formulates the problem of attacking black-box models as the perturbation to the victim's environment and tries to learn actions by the adversary that minimizes the victim model's reward.



[a] Nasser at. al , "Cross-Domain Transferability of Adversarial Perturbations", ( NeurIPS 2019 )
[b] Hamdi et. al, "SADA: Semantic Adversarial Diagnostic Attacks for Autonomous Applications" (AAAI 2020)



Minor issues :
The tables could have been illustrated as plots for easier following and comparisons

---

> ### Author Response · Authors · 2020-11-20
> **Response to AnonReviewer1**
>
> We would like to thank the reviewer for the positive comments, and we shall carefully revise the paper following your suggestions:
>
> * To the comment “...limited ablation study”: we appreciate the suggestion and we would like to respond as below:
>     * Selection process of $\beta$ values: the values 0 and 0.25 were tuned under a constraint of $\beta_1 < \beta_2$. We tried combinations of values in {0, 0.25, 0.5, 1.0} for $\beta_1$ and $\beta_2$ on MNIST and found that $\beta_1=0, \beta_2=0.25$ provided the best result on the validation set, and these values were then applied to all other experiments. We have added a section (Appendix D) in the revised paper to explain the selection process of $\beta$ values.
>     * Transferability between different victim models: thanks for this very insightful review. On CIFAR-10, we conducted an experiment (see Appendix E for details) to study the transferability of the policy network pre-trained on one victim model to attack several other victim models. When attacking a victim model using the policy network trained on a different victim model, we were able to obtain consistently better performance than HopSkipJumpAttack in the early stage of the attack. Therefore, in this transfer case, one can still benefit from our method, at least by just using the policy network to provide a better starting point for other attacks.
> * To the comment about the computational and memory efficiency, in our experiment when attacking a ResNet-18 model on ImageNet, for every 100 queries to the victim model, on average it costs our method extra 947ms on a single GPU (excluding the inference time of the victim model) to fine-tune the policy network, while HopSkipJumpAttack requires 354ms. Although it seems that our method is computational more intensive, the extra overhead is acceptable considering that the inference of the victim model is also costly (672ms), also the run-time of our method can be reduced by computation in parallel on multiple GPUs. As for GPU memory consumption, when using batch size 25, our method needs additional ~5GB GPU memory compared with HopSkipJumpAttack which needs ~2GB GPU memory. The computational and memory cost of our method can be reduced by compressing the policy network, which is to be explored in future work. We’ve added a section (Appendix F) in the revised paper to discuss the computation and memory complexity of our method.
> * To the comment about NATTACK and AutoZOOM, we would like to explain that the two methods target a different setting (the score-based black-box setting) from ours (the hard-label black-box setting), and it is indirect to adapt the two methods to our concerned setting. Although both NATTACK and AutoZOOM use deep auto-encoding architectures, their intentions are quite different from ours and it is also indirect to adapt our policy network into their algorithms. For NATTACK, it uses a DNN to directly predict the distribution of the adversarial example while our policy network only predicts the gradient direction at its input, thus it is unclear whether or not we can interpret the output of our policy network (i.e., the gradient) directly as an adversarial example like NATTACK. For AutoZOOM, the major purpose of bringing in an auto-encoder is dimensionality reduction. However, as our policy network calculates the gradient at its input, the inputs and outputs would always share the same dimensionalities. It is also hard to remove the “encoder part” from our policy network as in AutoZOOM, since the input image is a necessary component for the computational graph in our policy network.
> * We would like to thank the reviewer for the pointer to the relevant papers [a] [b]. We have added discussions about the two papers in the revised “related work” section in our paper.

---

> > ### Comment · AnonReviewer1 · 2020-11-20
> > **All concerns are addressed**
> >
> > The authors have addressed all of my concerns, and hence, I have increased the score of the paper to 7. This is a well-rounded paper with many aspects of the problem studied in the appendix. The work provides valuable insights to the community and therefore I recommend accepting the paper to ICLR'21.

---

### Official Review · AnonReviewer3 · 2020-11-07
**Interesting paper, clearly written, with good empirical result**

**Rating:** 7
**Confidence:** 3

**Review:**

This paper proposed a new black-box attack method in the hard-label setting. By using a well-designed policy network in a novel reinforcement learning formulation, the new method learns promising search directions of the adversarial examples and showed that query complexity is significantly reduced in experiments.

A couple questions I have:

How are \beta_1 and \beta_2 chosen?

Is there any convergence analysis? How do we guarantee the proposed attacking method will converge?

The proposed method is introduced as an attack that minimize L_2 distance.  Is it possible to extend this attacking method to L_inf?

I am curious, have you tried evaluate the attacking methods on DenseNet as the victim model?

Overall, I think this paper is very readable and is clearly written with a very good background and context. I found the idea of the paper original and interesting. And the authors have conducted experiments that show their new method has the best query efficiency, which is reasonable and aligns with their idea. For cons, this paper does not have a convergence analysis. And if the experiments could be conducted on more data sets and more victim models, then it would be more convincing.

---

> ### Author Response · Authors · 2020-11-20
> **Response to AnonReviewer3**
>
> We would like to thank the reviewer for the very insightful reviews. Below are our response to the comments:
>
> * To the comment “How are $\beta_1$ and $\beta_2$ chosen?”: the values 0 and 0.25 were tuned under a constraint of $\beta_1 < \beta_2$. We tried combinations of values in {0, 0.25, 0.5, 1.0} for $\beta_1$ and $\beta_2$ on MNIST and found that $\beta_1=0, \beta_2=0.25$ provided the best result on the validation set, and these values were then applied to all other experiments. We have added a section (Appendix D) in the revised paper to explain the selection process of $\beta$ values.
> * To the comment on convergence analysis, we would like to explain that, in practice, we can stop querying when no further distortion reduction can be obtained over the past several queries. In our implementation the baseline value $l_t$ is set to $0.05 * \delta$ if it goes below this value. As a result, once further distortion reduction is unavailable, all sampled actions would receive reward zero, which indicates no update on the parameters of the policy network either (according to REINFORCE). The theoretical convergence analysis of our method is entangled with DNN training dynamics which will be explored in future work.
> * To the comment “...extend this attacking method to $L_\infty$”: we think that it requires a different distribution assumption from Gaussian (as in our paper), and thus the extension can be nontrivial. In general, if a learnable Gaussian distribution is still applied, the sampled perturbations all become the same after being processed by a sign function. We will study it in more detail and propose a solution hopefully in future work.
> * To address the suggestion of introducing DenseNet as a victim model, we have conducted an experiment on CIFAR-10 to attack a DenseNet-100 victim model which has an error rate of 4.54% on the original test set (pre-trained weights: https://github.com/bearpaw/pytorch-classification). Below table reports the mean $\ell_2$ distortion on CIFAR-10 test images for untargeted attacks with different query budgets:
>
> |            Method            | @100 | @500 | @1K | @5K | @10K | @25K |
> |---|---|---|---|---|---|---|
> |    Boundary Attack    | 9.087 | 7.955 | 7.856| 3.057| 0.685 | 0.196 |
> | HopSkipJumpAttack | 2.655 | 0.845 | 0.472| 0.182| 0.141 | 0.117 |
> |               Ours              | 0.801 | 0.550 | 0.466| 0.247| 0.138 | 0.129 |

---

> > ### Comment · AnonReviewer3 · 2020-11-21
> > **Concerns and questions are addressed**
> >
> > I want to thank the authors for addressing my concerns and questions with great details, reasons and new experiments. The new experiment results look pretty good! After careful consideration, I would like to increase my score to 7.

---

### Decision · Program_Chairs · 2021-01-07
**Final Decision**

**Decision:**

Accept (Poster)

**Comment:**

The paper proposes an RL-based approach for decision-based attack. All the reviewers like the paper after the rebuttal phase, and we would like to encourage the authors to incorporate the new experiments in the camera-ready version. Furthermore, some recent decision-based attacks should also be included in the comparisons, such as

Li et al., QEBA: Query-Efficient Boundary-Based Blackbox Attack. (CVPR 2020)

Cheng et al., Sign-OPT: A Query-Efficient Hard-label Adversarial Attack. (ICLR 2020)